# Whole-Genome Sequencing Reveals Differences among *Kingella kingae* Strains from Carriers and Patients with Invasive Infections

Omer Murik,[a] David A. Zeevi,[a] Tzvia Mann,[a] Livnat Kashat,[b] Marc V. Assous,[b,c] Orli Megged,[c,d] Pablo Yagupsky[e]

[a]Translational Genomics Laboratory, Medical Genetics Institute, Shaare Zedek Medical Center, Jerusalem, Israel
[b]Microbiology Laboratory, Shaare Zedek Medical Center, Jerusalem, Israel
[c]Faculty of Medicine, Hebrew University of Jerusalem, Jerusalem, Israel
[d]Pediatric Department and Infectious Diseases Unit, Shaare Zedek Medical Center, Jerusalem, Israel
[e]Clinical Microbiology Laboratory, Ben-Gurion University of the Negev, Beer-Sheva, Israel

**ABSTRACT** As a result of the increasing use of sensitive nucleic acid amplification tests, *Kingella kingae* is being recognized as a common pathogen of early childhood, causing medical conditions ranging from asymptomatic oropharyngeal colonization to bacteremia, osteoarthritis, and life-threatening endocarditis. However, the genomic determinants associated with the different clinical outcomes are unknown. Employing whole-genome sequencing, we studied 125 international *K. kingae* isolates derived from 23 healthy carriers and 102 patients with invasive infections, including bacteremia ($n = 23$), osteoarthritis ($n = 61$), and endocarditis ($n = 18$). We compared their genomic structures and contents to identify genomic determinants associated with the different clinical conditions. The mean genome size of the strains was 2,024,228 bp, and the pangenome comprised 4,026 predicted genes, of which 1,460 (36.3%) were core genes shared by >99% of the isolates. No single gene discriminated between carried and invasive strains; however, 43 genes were significantly more frequent in invasive isolates, compared to asymptomatically carried organisms, and a few showed a significant differential distribution among isolates from skeletal system infections, bacteremia, and endocarditis. The gene encoding the iron-regulated protein FrpC was uniformly absent in all 18 endocarditis-associated strains but was present in one-third of other invasive isolates. Similar to other members of the *Neisseriaceae* family, the *K. kingae* differences in invasiveness and tropism for specific body tissues appear to depend on combinations of multiple virulence-associated determinants that are widely distributed throughout the genome. The potential role of the absence of the FrpC protein in the pathogenesis of endocardial invasion deserves further investigation.

**IMPORTANCE** The wide range of clinical severities exhibited by invasive *Kingella kingae* infections strongly suggests that isolates differ in their genomic contents, and strains associated with life-threatening endocarditis may harbor distinct genomic determinants that result in cardiac tropism and severe tissue damage. The results of the present study show that no single gene discriminated between asymptomatically carried isolates and invasive strains. However, 43 putative genes were significantly more frequent among invasive isolates than among pharyngeal colonizers. In addition, several genes displayed a significant differential distribution among isolates from bacteremia, skeletal system infections, and endocarditis, suggesting that the virulence and tissue tropism of *K. kingae* are multifactorial and polygenic, depending on changes in the allele content and genomic organization. Further analysis of these putative genes may identify genomic determinants of the invasiveness of *K. kingae* and its affinity for specific body tissues and potential targets for a future protective vaccine.

Address correspondence to Pablo Yagupsky, pyagupsky@gmail.com.

The authors declare no conflict of interest.

**KEYWORDS** *Kingella kingae*, carriage, invasive diseases, pangenome, virulence determinants, whole-genome sequencing

The increasing use of nucleic acid amplification assays has resulted in the recognition of *Kingella kingae* as the leading cause of osteoarthritis and a frequent agent of bacteremia in children 6 months to 4 years of age (1, 2). *Kingella kingae* also causes endocarditis in children and adults (3). Similar to other members of the *Neisseriaceae* family, *K. kingae* is asymptomatically carried on the oropharynx (4), and the colonized mucosa is the source of droplet transmission through close physical contact between young children and of the entry of *K. kingae* into the bloodstream, from which it disseminates to distant sites (5–7).

Skeletal system infections constitute 55% of all cases of invasive *K. kingae* disease, followed by occult bacteremia in 39% of cases, whereas endocarditis is uncommon, representing <2% of all clinical infections (2). *Kingella kingae* nonendocardial infections (NEIs) affect almost exclusively young children, are characterized by mild local and systemic inflammation, and, if adequately treated with antibiotics, have an excellent prognosis, leaving no long-term disabilities (1, 2, 8–10). In contrast, patients with *K. kingae* endocarditis have high fever and elevated acute inflammation markers (2, 10, 11). Despite the susceptibility of *K. kingae* to $\beta$-lactam and aminoglycoside antibiotics that are usually administered to endocarditis patients (12), this pathogen causes rapid destruction of the cardiac valves, and the resulting friable vegetations entail an increasing risk of life-threatening embolic phenomena in the brain and peripheral arteries (2, 3, 11). Patients with *K. kingae* endocarditis that is unresponsive to medical treatment require emergency cardiac surgery, the mortality rate is high, and many patients later need valve correction or replacement (2, 3, 11). The striking difference between the severe features of *K. kingae* endocarditis and the benign clinical course of NEI suggests that strains that invade the endocardium may harbor distinct genomic determinants that result in cardiac tropism, a more robust inflammatory response, and extensive tissue damage. However, despite the increasing appreciation of *K. kingae* as an important human pathogen, the species' population structure and the genomic determinants associated with the different clinical outcomes remain largely unknown. To the best of our knowledge, the full genomes of only 6 isolates have been published, and the draft assemblies of another 51 isolates have been deposited in public databases (https://www.ncbi.nlm.nih.gov/assembly). Employing whole-genome sequencing (WGS), a study was conducted to analyze the population structure of the species and to identify genes associated with asymptomatic carriage and invasive infections. The study results might contribute to identifying novel targets for a future protective vaccine.

## RESULTS

**Origin of the strains included in the study.** Twenty-three isolates from unrelated asymptomatic carriers, 23 from patients with bacteremia without focal infection, 61 from children with joint or bone infections, and 18 from children and adult patients with endocarditis were studied. Strains were isolated in Israel ($n = 91$), France ($n = 16$), Canada ($n = 8$), Spain ($n = 6$), Norway ($n = 2$), the United States ($n = 1$), and Russia ($n = 1$). A detailed description of these 125 strains is provided in Table 1.

**Genome size and gene content.** Each sequencing library resulted in 0.5 million to 2 million reads, giving a predicted average genome coverage of 30× to 150×. *De novo* assembly of the 125 genomes resulted in a mean genome size of 2,024,228 bp (95% confidence interval, 2,017,811 bp to 2,030,644 bp). The mean sequence length of the shortest contig at 50% of the total genome length ($N_{50}$) was 48.8 kb (range, 26 kb to 76 kb). Genome annotations of the *de novo* assemblies predicted an average of 1,964 putative genes per genome (range, 1,836 to 2,091 genes).

**Genomic diversity among the study's isolates.** Fifty-five distinct multilocus sequence typing (MLST) sequence types (STs) belonging to 9 ST complexes (STCs) were identified among the 125 sequenced isolates (Fig. 1A). Twenty-five isolates were less similar to other

**TABLE 1** Demographic, clinical, and MLST characteristics of the 125 *K. kingae* strains included in the study

| Isolate name | Isolation conditions | | | Clinical condition | | | | ST | STC |
|---|---|---|---|---|---|---|---|---|---|
| | Year | Country | City/region | Carriage (n = 23) | Skeletal infection (n = 61) | Bacteremia (n = 23) | Endocarditis (n = 18) | | |
| 1699 | 2014 | Israel | Jerusalem | | + | | | 71 | |
| 9001970 | 2012 | Israel | Center | | + | | | 24 | 23 |
| 1662-A6845 | 2015 | Israel | Jerusalem | | | | + | 24 | 23 |
| 35-A372 | 2010 | Israel | Jerusalem | | | | + | 21 | 23 |
| ATCC 23330 | 1966 | Norway | Oslo | + | | | | 1 | 1 |
| ATCC 23331 | 1960s | USA | Unknown | | | + | | 23 | 23 |
| ATCC 23332 | 1960s | Norway | Oslo | | | + | | 17 | 14 |
| AUD 31930140-S | 2013 | France | Paris | | + | | | 56 | 23 |
| B0458 | 2017 | Israel | Jerusalem | | + | | | 22 | 23 |
| B0605 | 2019 | Israel | Jerusalem | | + | | | 75 | |
| B0802 | 2016 | Israel | Jerusalem | | + | | | 72 | 14 |
| B0812 | 2019 | Israel | Jerusalem | | + | | | 6 | 6 |
| B10615 | 2008 | Israel | Center | | + | | | 11 | 11 |
| B1389 | 2018 | Israel | Jerusalem | | | + | | 72 | 14 |
| B1821 | 2015 | Israel | Jerusalem | | + | | | 6 | 6 |
| B3212 | 2018 | Israel | Jerusalem | | | | + | 25 | 23 |
| B3756 | 2017 | Israel | Jerusalem | | + | | | 75 | |
| B3892 | 2017 | Israel | Jerusalem | | + | | | 75 | |
| B4104 | 2019 | Israel | Jerusalem | | + | | | 73 | 14 |
| B5743 | 2019 | Israel | Jerusalem | | | + | | 73 | 14 |
| B6249 | 2021 | Israel | Jerusalem | | | | + | 75 | |
| B7142 | 2017 | Israel | Jerusalem | | + | | | 76 | |
| B7216 | 2018 | Israel | Jerusalem | | + | | | 73 | 14 |
| B7595 | 2016 | Israel | Jerusalem | | + | | | 6 | 6 |
| B7600 | 2018 | Israel | Jerusalem | | + | | | 71 | |
| B8255 | 2019 | Israel | Jerusalem | | | + | | 21 | 23 |
| B8907 | 2018 | Israel | Jerusalem | | | | + | 73 | 14 |
| B9852 | 2017 | Israel | Jerusalem | | + | | | 75 | |
| BB11960 | 2004 | Israel | Center | | + | | | 35 | 35 |
| BOU 30672 | 2010 | France | Paris | | + | | | 14 | 14 |
| CA105 | 2009 | Spain | Catalonia | | | + | | 62 | 23 |
| CA138 | 2012 | Spain | Catalonia | | + | | | 14 | 14 |
| CA139 | 2012 | Spain | Catalonia | + | | | | 63 | |
| CA20 | 1998 | Spain | Catalonia | | + | | | 23 | 23 |
| CA64 | 2004 | Spain | Catalonia | | | + | | 6 | 6 |
| CA77 | 2006 | Spain | Catalonia | | + | | | 25 | 23 |
| CA99 | 2009 | Spain | Catalonia | | + | | | 61 | 6 |
| CAN1 | 2003 | Canada | Montreal | | + | | | 25 | 23 |
| CAN16 | 2010 | Canada | Montreal | | + | | | 58 | 35 |
| CAN2 | 2005 | Canada | Montreal | | + | | | 12 | 11 |
| CAN21 | 2012 | Canada | Montreal | | + | | | 59 | 6 |
| CAN22 | 2012 | Canada | Unknown | | | + | | 6 | 6 |
| CAN25 | 2013 | Canada | Montreal | | + | | | 60 | 14 |
| CAN8 | 2007 | Canada | Montreal | | + | | | 14 | 14 |
| CAN9 | 2007 | Canada | Montreal | | + | | | 62 | 23 |
| CIP 101722 | 1985 | France | Grenoble | | | + | | 14 | 14 |
| CIP 102473 | 1986 | France | Paris | | + | | | 19 | |
| CIP 73.01 | 1972 | France | Unknown | | | + | | 14 | 14 |
| DAG 31560001-S | 2013 | France | Paris | | + | | | 55 | 23 |
| DER 112012-1 | 2012 | France | Paris | | + | | | 42 | 14 |
| ETI 126580 | 2011 | France | Paris | | + | | | 43 | |
| F0228 | 2019 | Israel | Jerusalem | | + | | | 75 | |
| F1990_B1887 | 2017 | Israel | Jerusalem | | + | | | 6 | 6 |
| FOF 3022006-S | 2013 | France | Paris | | + | | | 46 | 6 |
| KK100 | 1996 | Israel | South | | + | | | 24 | 23 |
| KK104 | 1994 | Israel | South | + | | | | 23 | 23 |
| KK113 | 1996 | Israel | South | + | | | | 31 | 34 |
| KK114 | 1996 | Israel | South | + | | | | 23 | 23 |
| KK12 | 1994 | Israel | South | + | | | | 14 | 14 |
| KK120 | 1996 | Israel | South | + | | | | 11 | 11 |

**TABLE 1** (Continued)

| | Isolation conditions | | | Clinical condition | | | | | |
|---|---|---|---|---|---|---|---|---|---|
| Isolate name | Year | Country | City/region | Carriage (*n* = 23) | Skeletal infection (*n* = 61) | Bacteremia (*n* = 23) | Endocarditis (*n* = 18) | ST | STC |
| KK128 | 1997 | Israel | North | | | | + | 24 | 23 |
| KK136 | 1998 | Israel | North | | | + | | 14 | 14 |
| KK138 | 1998 | Israel | North | | + | | | 24 | 23 |
| KK144 | 1998 | Israel | South | | + | | | 24 | 23 |
| KK145 | 1998 | Israel | Jerusalem | | + | | | 6 | 6 |
| KK153 | 1999 | Israel | North | | | | + | 23 | 23 |
| KK154 | 1999 | Israel | South | | | + | | 18 | 14 |
| KK164 | 2000 | Israel | South | | | + | | 23 | 23 |
| KK168 | 2001 | Israel | South | | + | | | 48 | |
| KK171 | 2001 | Israel | North | | + | | | 5 | |
| KK174 | 1998 | Israel | South | | | + | | 29 | 29 |
| KK180 | 2002 | Israel | Center | | | | + | 10 | 35 |
| KK183 | 2002 | Israel | North | | + | | | 13 | |
| KK189 | 2002 | Israel | South | | + | | | 35 | 35 |
| KK190 | 2002 | Israel | South | | | | + | 24 | 23 |
| KK194 | 2003 | Russia | St. Petersburg | + | | | | 9 | 29 |
| KK197 | 2003 | Israel | Center | | | | + | 24 | 23 |
| KK199 | 2004 | Israel | Jerusalem | | | | + | 6 | 6 |
| KK208 | 2004 | Israel | South | | | + | | 27 | |
| KK212 | 2004 | Israel | South | + | | | | 24 | 23 |
| KK220 | 2004 | Israel | South | | + | | | 51 | |
| KK242 | 2005 | Israel | South | | | + | | 27 | |
| KK244 | 2005 | Israel | North | | | + | | 7 | 6 |
| KK245 | 2005 | Israel | South | | | + | | 33 | 34 |
| KK247 | 2006 | Israel | Jerusalem | | | | + | 38 | 34 |
| KK253 | 2007 | Israel | South | | + | | | 23 | 23 |
| KK263 | 2008 | Israel | Center | | | + | | 7 | 6 |
| KK267 | 2008 | Israel | South | | + | | | 53 | |
| KK3 | 1994 | Israel | South | + | | | | 5 | |
| KK411 | 2008 | Israel | Jerusalem | | | | + | 6 | 6 |
| KK416 | 2011 | Israel | Jerusalem | | | | + | 24 | 23 |
| KK420 | 2011 | Israel | Center | | + | | | 54 | |
| KK433 | 2012 | Israel | Center | | + | | | 52 | 35 |
| KK444 | 2013 | Israel | Center | | | | + | 12 | 11 |
| KK448 | 2014 | Israel | North | | | | + | 74 | |
| KK470 | 2018 | Israel | South | | | | + | 35 | 35 |
| KK56 | 1994 | Israel | South | | + | | | 11 | 11 |
| KK60 | 1994 | Israel | South | | | | + | 2 | 1 |
| KK70 | 1993 | Israel | South | | + | | | 4 | 14 |
| KK81 | 1991 | Israel | South | | + | | | 22 | 23 |
| KK83 | 1991 | Israel | South | | + | | | 14 | 14 |
| KK86 | 1994 | Israel | South | + | | | | 6 | 6 |
| KK88 | 1996 | Israel | North | | + | | | 2 | 1 |
| KK92 | 1996 | Israel | North | | | + | | 6 | 6 |
| KK93 | 1995 | Israel | South | | | + | | 24 | 23 |
| KK98 | 1992 | Israel | South | | | + | | 22 | 23 |
| KWG1 | 2013 | France | Paris | | + | | | 41 | 14 |
| MAR_1853 | 2011 | France | Paris | | + | | | 32 | 34 |
| N10-10770 | 2010 | France | Nantes | | | + | | 28 | 29 |
| N10-6318 | 2010 | France | Nantes | | + | | | 23 | 23 |
| NICE476 | 2012 | France | Nice | | + | | | 6 | 6 |
| PER1851 | 2011 | Israel | South | + | | | | 31 | 34 |
| PER251 | 2010 | Israel | South | + | | | | 58 | 35 |
| PER2748 | 2011 | Israel | South | + | | | | 28 | 29 |
| PER3 | 2010 | Israel | South | + | | | | 8 | |
| POH14284 | 2011 | France | Paris | | + | | | 18 | 14 |
| SAI11985 | 2011 | France | Paris | | + | | | 26 | 23 |
| Sch1614 | 2012 | Israel | South | + | | | | 27 | |
| Sch187 | 2010 | Israel | South | + | | | | 3 | 3 |
| Sch1931 | 2012 | Israel | South | + | | | | 35 | 35 |

**TABLE 1** (Continued)

| | Isolation conditions | | | Clinical condition | | | | | |
|---|---|---|---|---|---|---|---|---|---|
| | | | | Carriage | Skeletal infection | Bacteremia | Endocarditis | | |
| Isolate name | Year | Country | City/region | (*n* = 23) | (*n* = 61) | (*n* = 23) | (*n* = 18) | ST | STC |
| Sch2108 | 2012 | Israel | South | + | | | | 24 | 23 |
| Sch258 | 2010 | Israel | South | + | | | | 18 | 14 |
| Sch429 | 2011 | Israel | South | + | | | | 11 | 11 |
| Sch87 | 2010 | Israel | South | + | | | | 49 | |
| ZUL30220039-S | 2013 | France | Paris | + | | | | 50 | 1 |
| Total | | | | 23 | 61 | 23 | 18 | | |

isolates and could not be allocated to a specific STC. The most frequent STs were ST-24 (*n* = 12 isolates), ST-6 (*n* = 12), ST-14 (*n* = 8), ST-23 (*n* = 8), ST-75 (*n* = 6), ST-11 (*n* = 4), ST-73 (*n* = 4), and ST-35 (*n* = 4) (Table 1). New STs were assigned for 15 isolates because their allele content did not match any of the STs deposited in the *K. kingae* typing data set (http://bigsdb.pasteur.fr/kingella). The novel STs included ST-71 (2 isolates), which belonged to STC-13, ST-72 (*n* = 2) and ST-73 (*n* = 4), both of which belonged to STC-14, and ST-75 (*n* = 6) and ST-76 (*n* = 1), which could not be assigned to a STC. These 15 STs were restricted to the area of Jerusalem, Israel, and were isolated between 2017 and 2021 (Table 1).

To use the power of WGS to increase the genomic diversity resolution, compared to standard MLST, we performed core genome MLST (cgMLST) for the isolates. Allelic diversity for 1,429 single-copy genes shared by all isolates was assessed, resulting in 125 different cgMLST types (Fig. 1B); in order to organize the cgMLST and MLST phyloge-

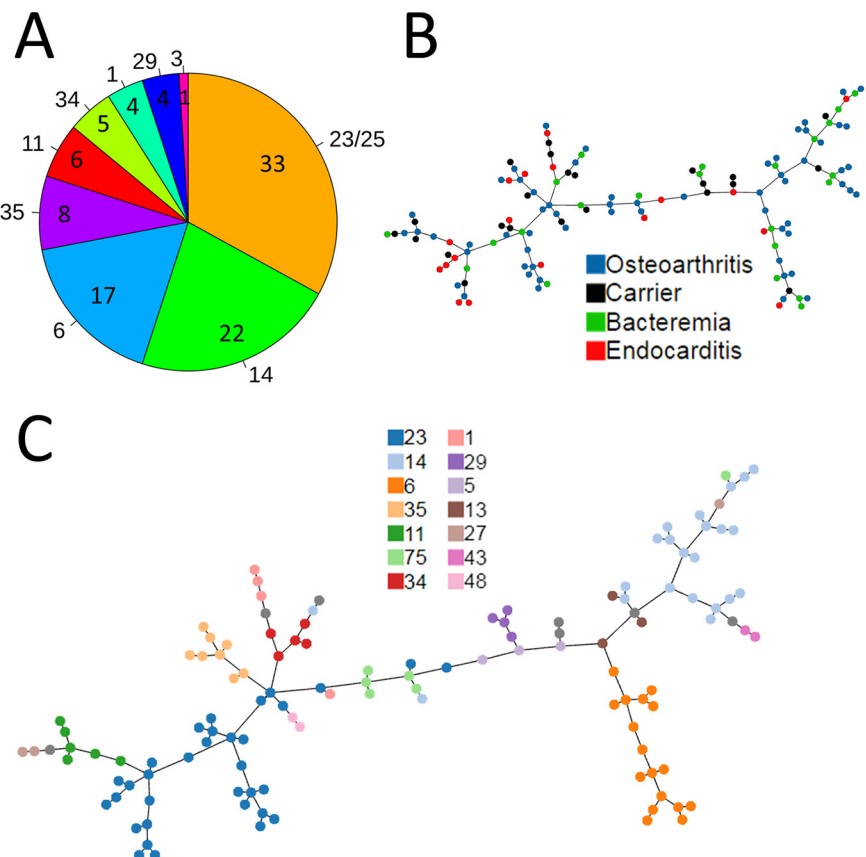

**FIG 1** Diversity of the 125 *K. kingae* isolates analyzed in this study. (A) Pie chart based on the number of isolates belonging to the indicated MLST STC. (B) PHYLOVIZ visualization of the cgMLST of all 125 isolates, color-coded according to the clinical condition. (C) Same as in panel B, color-coded for MLST.

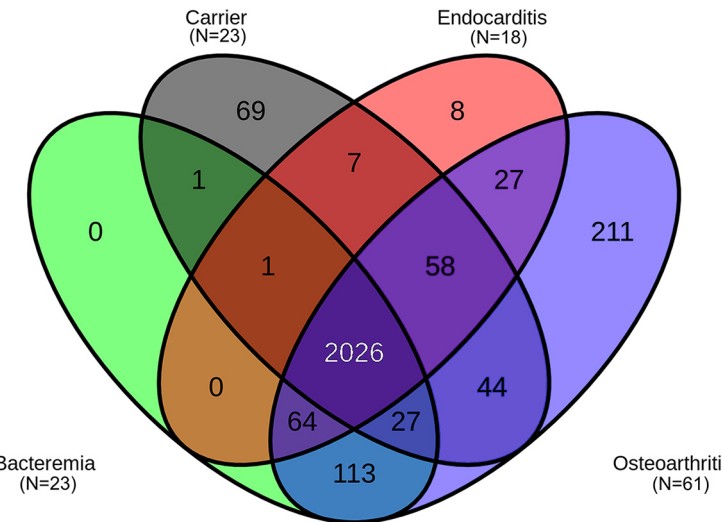

**FIG 2** Venn diagram showing the number of genes in the pangenome associated with each of the four clinical conditions, including only genes found in ≥5 isolates.

nies, we color-coded the same scheme according to the MLST ST of each isolate (Fig. 1C). Both MLST and cgMLST revealed that the isolates did not cluster based on the clinical condition. This means that, for example, there was no common phylogenetic ancestor for all isolates cultured from endocarditis patients. Because the visualization of cgMLST results (Fig. 1B and C) is based on allelic differences and lacks information about the number of sequence differences between the genomes, we further performed phylogenetic analysis of the core genome. We used Roary, MAFFT, and FastTree (see Materials and Methods) to produce the core genome maximum likelihood unrooted phylogenetic tree (see Fig. S1 in the supplemental material). The phylogenetic analysis supported the cgMLST results, demonstrating that the ability to infect the endocardium is not related to a specific evolutionary branch. Pairwise analysis of single-nucleotide variations (SNVs) between every two genome assemblies revealed 5 clusters of isolates that had <50 SNV differences, with 2 to 5 isolates in each cluster (see Fig. S2). Since this threshold was never tested in *K. kingae*, we chose the cutoff value of 50 SNVs based on studies with other bacteria (13). This analysis revealed that our isolates were very diverse and there was no group of clonal isolates that would bias our downstream analyses. Moreover, we observed clusters of high genetic relatedness among isolates cultured from patients with different clinical conditions.

**Core genome and pangenome.** Comparative genomic analysis of all 125 genome assemblies revealed a set of 1,460 core genes detected in all isolates, 85 soft-core genes found in 95 to 99% of the isolates, 814 shell genes found in 15 to 95% of the isolates, and 1,667 cloud genes found in <15% of the isolates. In total, the pangenome consisted of 4,026 predicted genes. Of these, 3,430 (85%) have orthologues according to eggNOG-mapper analysis. As expected, the number of core genes gradually diminished as an increasing number of genomes were studied. However, the value stabilized, and adding more genomes did not result in significant reductions of the core and soft-core sizes (see Fig. S3). To further validate this point, we added the genome assemblies of 47 *K. kingae* strains deposited in the National Center for Biotechnology Information (NCBI) database. The addition of these genomes, which represented a 37.9% increment in the strain population, increased the pangenome size to 5,393 genes (a 1,367-gene [33.9%] increment) but only reduced the core genome size to 1,393 genes (a decrease of merely 67 genes [4.5%]), indicating that the species exhibits a high degree of genomic conservation.

**Functional genomics of the pangenome.** The presence or absence of the pangenome genes according to the clinical condition is shown in Fig. 2 for genes present in ≥5 isolates. As expected, the vast majority of the genes (*n* = 2,026) were shared.

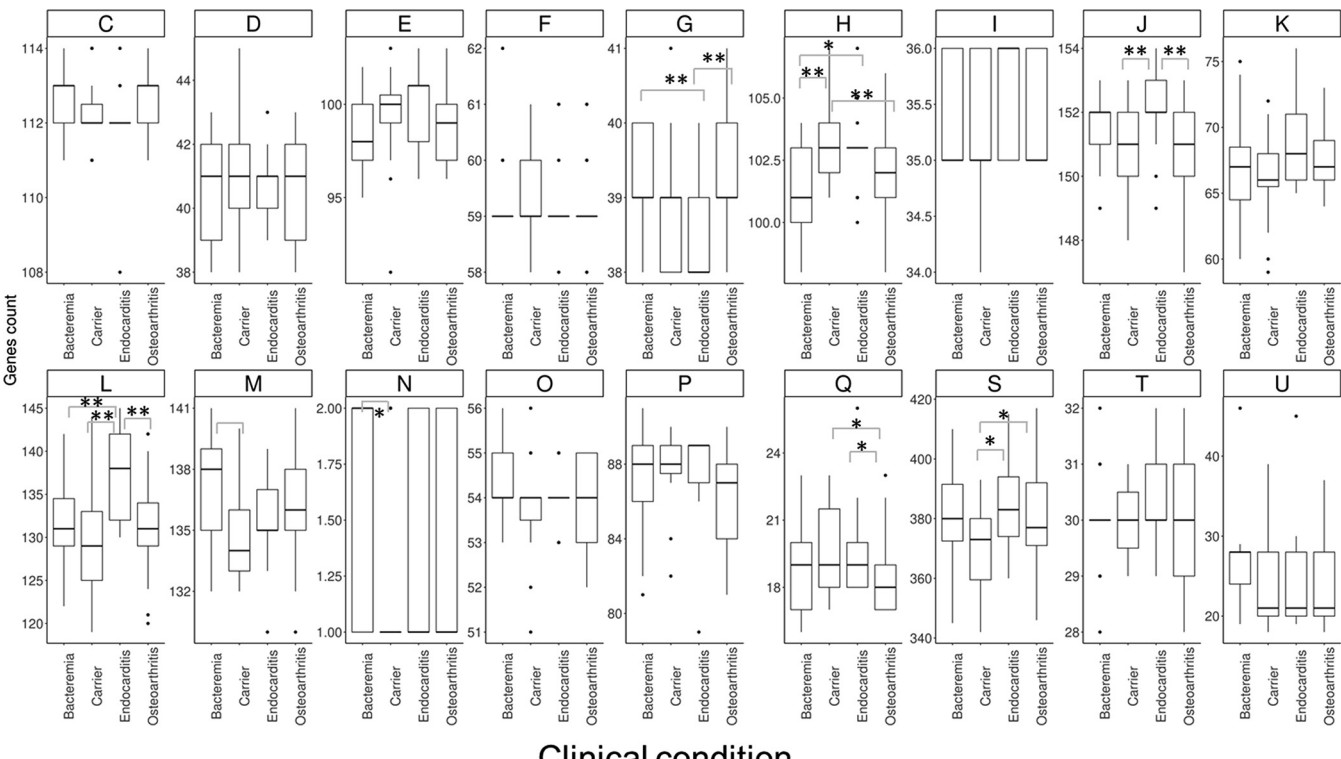

**FIG 3** Distribution of the number of genes per isolate associated with each COG category and distributed in the four clinical condition groups. *, $P < 0.05$; **, $P < 0.01$, one-way ANOVA. COG categories are as follows: C, energy production and conversion; D, cell cycle control and mitosis; E, amino acid metabolism and transport; F, nucleotide metabolism and transport; G, carbohydrate metabolism and transport; H, coenzyme metabolism; I, lipid metabolism; J, translation; K, transcription; L, replication and repair; M, cell wall/membrane/envelope biogenesis; N, cell motility; O, posttranslational modification, protein turnover, and chaperone functions; P, inorganic ion transport and metabolism; Q, secondary structure; T, signal transduction; U, intracellular trafficking and secretion; S, unknown function.

However, several genes were particularly associated with specific conditions. For instance, 8 genes were found in ≥5 of the 18 endocarditis-associated isolates but in ≤4 isolates from the 107 patients with NEI and healthy carriers.

The functional group for each protein encoded in the pangenome was determined using the database of the Clusters of Orthologous Genes (COGs). We further investigated the number of genes in each COG for each isolate and grouped the isolates according to their clinical condition (Fig. 3). The numbers of proteins per isolate in COGs G, H, J, L, N, Q, and S (carbohydrate metabolism and transport, coenzyme metabolism, translation, replication and repair, cell motility, secondary structure, and unknown function, respectively) showed statistically significant differences between isolates belonging to the four clinical conditions (one-way analysis of variance [ANOVA] results) (Fig. 3). Interestingly, the mean number of genes from COG L (replication and repair) was significantly higher in endocarditis-associated isolates.

**Genes associated with invasive infections.** To look for genomic features that facilitate deep tissue invasion, we searched for genes that were present in all invasive isolates and absent in colonizers and vice versa. Although no single gene met this criterion, 43 genes were more common among invasive isolates than among carried organisms (see Table S1). None of the genes was significantly enriched after multiple-test correction (Bonferroni-adjusted $P$ of <0.05). While 7 of these genes were components of type IV secretion systems and 2 others were related to pilus assembly, the rest were mostly for proteins of unknown function.

**Endocarditis-associated genes.** We could not identify genes whose presence was exclusively associated with bacterial endocarditis. However, 13 genes were found in ≥3 of the 18 endocarditis-causing isolates and were lacking in NEI organisms. Six of those genes are of unknown function, 3 are for transposases, and 2 are for DNA methylases; the

remaining 2 are for a signal transduction protein containing a PAS/PAC domain and a 5-oxoprolinase (*pxpB*). Another 14 genes were more common among endocarditis-associated strains than among NEI-associated or carried strains (see Table S2). Six of those genes were significantly enriched after multiple-test correction (Bonferroni-adjusted *P* value of <0.05). Remarkably, the pangenome includes two types of the *frpC* gene, encoding the iron-regulated protein FrpC, one of which was absent in all endocarditis-associated strains and present in 31 (29.0%) of the 107 isolates from carriers and NEI patients and in 27 (32.1%) of the 84 isolates derived from bacteremia or skeletal system infections (*P* values of <0.05 for both comparisons). Further investigation revealed another type of the *frpC* gene that was found in isolates from all clinical conditions.

**Comparison of endocarditis-associated and NEI-associated strains belonging to STC-23/25.** Remarkably, 8 (44.4%) of the 18 endocarditis-associated isolates belonged to STC-23/25, compared to only 25 (23.4%) of 107 non-endocarditis-associated strains (*P* = 0.112 by the chi-square test). Therefore, we looked for genomic differences between endocarditis-associated and non-endocarditis-associated strains within this STC. No particular genes were found in all endocarditis-associated isolates and absent in those derived from patients with other conditions, although 9 genes were overrepresented in STC-23/25 endocarditis-associated isolates, all of them of unknown function (see Table S3). None of them remained significantly enriched after multiple-test correction (Bonferroni-adjusted *P* value of <0.05).

## DISCUSSION

Although the number of healthy carriers of pathogenic bacteria is much larger than the number of individuals with disease, research on the population structure of many bacteria has been hampered by strain collections that are biased toward particularly virulent isolates, often neglecting less invasive organisms and underestimating the extent of genetic diversity of the species (14). In the present study, we selected a large international sample of colonizing and invasive *K. kingae* strains isolated over a long period to ensure representative genomic diversity. The results show that the genomic content of *K. kingae* (~2.2 million bp) is in the range of the sizes of other pathogenic *Neisseria* species, such as *Neisseria meningitidis* and *Neisseria gonorrhoeae* (https://www.ncbi.nlm.nih.gov/datasets/genome/?taxon=482).

*Kingella kingae* is naturally transformable, and horizontal gene transfer explains its remarkable genomic heterogeneity (15, 16). To date, 70 distinct MLST profiles have been described for the species (2, 16), and the results of the present study increase the number of known allele combinations to 76. The strain population included many STs belonging to the globally distributed STCs STC-6, -14, -23, -25, and -35 but also STs exhibiting a more restricted geographic dispersion (15, 16) and 5 novel STs limited to the Jerusalem area.

The fact that a given strain was isolated from the oropharynx does not necessarily imply that the organism is unable to cause an invasive infection. All of the strains included in the study had initially colonized the oropharynx, although only those isolated from individuals with disease had also breached the epithelium and entered the bloodstream, causing bacteremia. Of those, a few subsequently invaded the endocardial or skeletal tissues, for which the species exhibits particular affinity. Therefore, although mucosal colonization is a *sine qua non* precondition for the development of clinical disease, not all *K. kingae* strains are able to penetrate the oropharyngeal epithelial barrier, survive in the bloodstream, and colonize the skeletal system or the endocardium. Previously, a total of 32 distinct *K. kingae* clones were identified by pulsed-field gel electrophoresis (PFGE) (2), of which 5 (clones B, H, K, N, and P) were isolated in 132 (72.9%) of 181 Israeli patients with disease, indicating increased virulence (17). Remarkably, clones A, C, G, J, M, R, and T, which collectively represented 93 (38.8%) of all pharyngeal isolates from 240 healthy carriers (5), were detected in only 10 (4.4%) invasive infections, suggesting diminished invasive capability (17). Wide differences in virulence among *K. kingae* strains were also demonstrated in animal models, ranging from an inability to establish infection to a rapidly progressing and fatal illness (16, 18). In the present study, the carried group of isolates probably represented a mixture of

organisms with diverse invasive capabilities. This heterogeneity might have blurred the genomic differences between true mere colonizers (i.e., colonizing strains unable to cause disease) and invasive organisms. Despite this potential attenuation, the comparisons demonstrated genomic differences between organisms isolated from the oropharyngeal mucosa and those detected in infected body sites.

The possibility that *K. kingae* strains also show affinity for specific human tissues is supported by observations of outbreaks of infection in daycare centers (19). These events are naturally occurring quasiexperiments in which a single strain is introduced into a crowded facility attended by a susceptible young population, simultaneously or successively infecting multiple children within a short period (19). This phenomenon offers a unique opportunity to observe the strain's tropism for specific body sites. Analysis of these clusters has demonstrated that some *K. kingae* strains tend to cause the same clinical disease, i.e., septic arthritis, osteomyelitis, or tenosynovitis, suggesting tissue specificity, while others invade a variety of body niches, including the endocardium (19–21).

The present study showed that the ability to invade the cardiac valves is widespread among *K. kingae* strains, and 13 different STs were identified among the 18 endocarditis-associated isolates. However, 8 of the identified STs belonged to the close-knit STC-23/25. It should be pointed out that members of this STC, i.e., strain SAN 38360 (22) and strain COU 1310053120 (15), were recovered from two French children with bacterial endocarditis. STC-23/25 represented 52 (26.5%) of 196 non-endocarditis-associated strains in an international collection of invasive organisms isolated between 1966 and 2014, indicating that this STC is highly virulent, has disseminated worldwide, and remained stable over a long period (15, 16). However, the fraction of cases of endocarditis caused by STC-23/25 organisms in the present study (8 [44.4%] of 18 cases) did not statistically differ from the representation of this STC among NEI strains (15, 16) ($P = 0.18$), suggesting that this STC, although highly invasive, does not exhibit a particular tropism toward the cardiac valves.

Unlike *Enterobacterales* species, among which acquisition of pathogenic islands by horizontal transfer results in strains that cause distinct clinical diseases (23), the genetics of pathogenicity in the *Neisseriaceae* family are more complex, depending on combinations of multiple virulence-associated genes distributed throughout the genome (24). The present study results failed to identify single genes whose presence or absence could clearly separate mere colonizers from invasive strains. Instead, *K. kingae* organisms associated with bacteremia, bone and joint disease, and endocarditis showed statistically significant differences in the distribution of multiple genes. The observed differences in virulence and tissue tropism between *K. kingae* isolates may then be multifactorial and polygenic, depending on subtle changes in the allele content and the overall genomic organization (24). Alternatively, it can be argued that the aggressive clinical course observed among patients with *K. kingae* endocarditis depends not on the biological features of the strain but on the protective microenvironment of cardiac valve vegetations, which is poorly accessible to antibiotics, neutrophils, antibodies, and other host defense molecules (25). This inaccessibility results in bacterial multiplication to a very high density, with consequent tissue destruction, and the disintegration of the friable vegetations causes persistent bacteremia and metastatic embolic phenomena (25).

It should be pointed out, however, that a gene encoding the iron-regulated protein FrpC was uniformly absent in all 18 endocarditis-associated strains but was present in one-third of NEI-associated isolates (another type of FrpC was found in all isolates). Although the role of FrpC in the pathogenesis of *K. kingae* infections has not yet been determined, the analogous protein of *N. meningitidis* appears to function as an adhesin to the respiratory epithelium (26). The *frpC* gene is present in all invasive meningococcal strains and the majority of carried meningococcal strains (27), while the species is an exceptional etiology of bacterial endocarditis (28). These observations could lead to the speculation that the *K. kingae* FrpC protein offers an initial selective advantage, facilitating the colonization of the pharyngeal mucosa, but is detrimental to the

invasion of endocardial tissues. Future research is needed to elucidate the function of *K. kingae* genes that exhibit differential distribution among the associated clinical conditions and, particularly, the potential role of the absence of the *frpC* gene in the transition from asymptomatic carriage to bacterial endocarditis.

No cluster of genes was found to be statistically enriched in the genomes of isolates that caused invasive infection (see Table S1 in the supplemental material). This again supports the possibility that the result of *K. kingae* infection is not determined by bacterial genetics. However, six genes were statistically enriched in endocarditis-causing strains (see Table S2). It should be mentioned that this list must be carefully examined, because some genes are enriched only as a result of the clustering algorithm and actually are represented in isolates from all conditions. Another point is that none of the genes on this list is found exclusively in all endocarditis-causing isolates, and thus the tissue tropism cannot be fully explained by genetic factors. The functional analysis of five of these genes does not point to any of them having a function that may explain their distribution, such as functions related to virulence, tissue preferences, or biofilm formation. Only *icmT*, an isoprenylcysteine carboxyl methyltransferase, was shown to have an essential role in host cell pore formation, as mutants of this gene in *Legionella pneumophila* were affected regarding host cell lysis (29). Further investigation of molecular mechanisms and sequencing of additional clinical *K. kingae* isolate genomes may shed more light on the roles of these genes.

*Kingella kingae* is a common pathogen of early childhood, causing a variety of invasive diseases affecting the skeletal system and the endocardium. The present study results might contribute to identifying novel targets for a future protective vaccine.

## MATERIALS AND METHODS

**Source of the study strains.** The clinical microbiology laboratory of the Soroka University Medical Center in southern Israel has been a pioneer in the research on *K. kingae* and its diseases (30) and, over the past 3 decades, has isolated the organism from >150 infected patients and >500 asymptomatic carriers. In addition, the laboratory has gathered invasive *K. kingae* isolates from other regions of the country and international strains from Europe and North America (15, 16, 31). *K. kingae* strains derived from patients with endocarditis and other invasive infections that had been collected at the Shaare Zedek Medical Center in Jerusalem, Israel, were also available for the study.

To achieve a representative sample of the *K. kingae* population, a strain collection was assembled to include strains with a wide variety of PFGE STs that had been isolated between the 1960s and 2021 from individuals with a variety of invasive infections living in different geographic locations, as well as oropharyngeal strains from epidemiologically unrelated carriers (15, 16, 31). All strains have been kept frozen at −80°C in 15% glycerol-containing Mueller-Hinton medium since isolation.

**Clinical conditions associated with the *K. kingae* isolates.** For the purposes of the data analysis, the clinical conditions associated with the isolates were classified into two broad categories, namely, asymptomatic oropharyngeal carriage and invasive diseases. The latter group comprised bacteremia with no focal infection (occult bacteremia), skeletal system infections (including septic arthritis and osteomyelitis), and bacterial endocarditis. Isolates recovered from blood cultures drawn from patients with focal diseases such as septic arthritis or endocarditis were allocated to the skeletal system or the endocarditis category, respectively, and not to the bacteremia group.

**DNA extraction and WGS.** Genomic DNA was extracted from pure cultures using a DNeasy tissue kit (Qiagen, Germantown, MD) according to the manufacturer's instructions. DNA libraries were prepared using the Nextera XT kit (Illumina, San Diego, CA) according to the manufacturer's protocol and were sequenced with 150-bp single-end reads using an Illumina NextSeq platform.

**Bioinformatic analysis.** Raw sequencing reads were trimmed and filtered using Cutadapt, and filtered reads were used for *de novo* assembly with SPAdes v3.13.1 (32). Assemblies were evaluated with QUAST (33). Sequencing and assembly statistics are described in Table S4 in the supplemental material. Each isolate's assembly was checked for contamination using Kraken v2 (34), and genes were predicted and annotated using Prokka v1.14.6 (35). The presence of plasmids, antimicrobial resistance genes, and virulence genes was predicted using the PlasmidFinder, ResFinder, and Virulence Factor Database (VFDB) databases, respectively.

Pangenomes and core genomes were determined using Roary v3.13.0 (36). Phylogenetic relationships between genomes were assessed by aligning the core genome sequences using MAFFT v7.475 (37), and approximately maximum likelihood phylogenetic trees were built using FastTree v2.1.11 (38), with 1,000 resamples of the Shimodaira-Hasegawa test. MLST of all isolates was performed with the *de novo* assemblies using MLST software (https://cge.food.dtu.dk/services/MLST/). Isolates exhibiting previously undescribed allele combinations were given new ST numbers and deposited in the Institut Pasteur database (http://bigsdb.pasteur.fr/kingella). cgMLST was performed using chewBBACA (39).

**STCs.** Isolates were grouped into STCs if they differed at no more than 1 locus from at least one other member of the group. Founder genotypes of STCs were defined as the ST of the STC with the highest number of neighboring STs (single-locus variants). Early in the research, it became apparent that STC-23 and STC-25 were overrepresented among endocarditis-associated strains. The founder ST of STC-23 and that of STC-25 are closely related, differing at only 2 loci and, for the purposes of the analysis, we gathered these two STCs into a single STC and named it STC-23/25. The resulting STC-23/25 included ST-21, -22, -23, -24, -25, -26, -44, -55, -56, and -62. To compare the genomic content of STC-23/25 endocarditis-associated strains with that of strains derived from individuals with other conditions, the study population was enriched with additional STC-23/25 organisms isolated from healthy carriers, patients with bacteremia, and children with skeletal infections.

**Statistical analysis.** Gene contents in isolates related to the different clinical conditions were compared using the chi-square test, and *P* values were adjusted for multiple tests with the Bonferroni correction. *P* values of <0.05 were considered significant for all calculations. One-way ANOVA was employed to find genes whose distribution showed significant differences between the various clinical condition groups. Statistical analysis and data visualization were conducted in R v4.0.3 using the ggplot2 and pheatmap packages.

**Ethics approval.** Ethics approval was obtained from the local institutional review board (Helsinki Ethics Committee) (approval number 0511-20-SZMC).

**Data availability.** The raw sequencing data for *K. kingae* isolates reported in this study have been deposited in the NCBI Sequence Read Archive (SRA) under BioProject accession number PRJNA891445. The genome assemblies have been deposited in the BIGSdb-Pasteur database (https://bigsdb.pasteur.fr/kingella).

## SUPPLEMENTAL MATERIAL

Supplemental material is available online only.

**SUPPLEMENTAL FILE 1**, DOCX file, 0.02 MB.
**SUPPLEMENTAL FILE 2**, DOCX file, 0.02 MB.
**SUPPLEMENTAL FILE 3**, DOCX file, 0.02 MB.
**SUPPLEMENTAL FILE 4**, DOCX file, 0.03 MB.
**SUPPLEMENTAL FILE 5**, PDF file, 0.6 MB.

## ACKNOWLEDGMENTS

No external financial support was received for this work.

We do not have a commercial or other association that might pose a conflict of interest.

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
