## [Reviewer comments · Microbiology Spectrum]

Microbiology Spectrum

Whole Genomic Sequencing Reveals Differences among *Kingella kingae* Strains from Carriers and Patients with Invasive Infections

Omer Murik, David Zeevi, Tzvia Mann, Livnat Kashat, Marc Assous, Orli Megged, and Pablo Yagupsky

Corresponding Author(s): Pablo Yagupsky, Ben-Gurion University of the Negev

Review Timeline:

Submission Date:	October 25, 2022
Editorial Decision:	February 15, 2023
Revision Received:	April 16, 2023
Accepted:	April 26, 2023

Editor: Po-Yu Liu

Reviewer(s): Disclosure of reviewer identity is with reference to reviewer comments included in decision letter(s). The following individuals involved in review of your submission have agreed to reveal their identity: Yao-Ting Huang (Reviewer #1)

Transaction Report:

DOI: <https://doi.org/10.1128/spectrum.03895-22>

February 15, 2023

Prof. Pablo Yagupsky
Ben-Gurion University of the Negev
Clinical Microbiology Laboratory
Shderot Reger
Beer-Sheva
Israel

Re: Spectrum03895-22 (Whole Genomic Sequencing Reveals Differences among *Kingella kingae* Strains from Carriers and Patients with Invasive Infections)

Dear Prof. Pablo Yagupsky:

Link Not Available

Sincerely,

Po-Yu Liu

Journals Department
Reviewer comments:

Reviewer #1 (Comments for the Author):

Murisk et al. sequenced 125 *Kingella kingae* isolates associated with bacteremia (n=23), osteoarthritis (n=61), and endocarditis (n=18). The authors identified 43 genes possibly discriminative of healthy and invasion, as well as others associated with skeletal system infections, bacteremia, and endocarditis. The FrpC was absent in all endocarditis samples but present in one-third of other invasive samples. In all, the authors provide rich genomic material for further research on *Kingella kingae* infections. However, the manuscript still needs some clarifications on both the results and methods.

1. It looks to me Figure 1(B) and (C) redundantly plotted the cgMLST (or core genes) phylogeny of the 125 isolates just by two

distinct software. This confused the audience as it provided no new messages. Instead, I would suggest comparing the cgMLST with the MLST phylogenies, which may at least see the strength and weaknesses of each method.

2. It's difficult to connect the phylogenetic analysis in the results with the methods. It looks like there are three cgMLST-like analyses performed (L137-151). But only roary and chewBBACA appeared in the methods. And it's hard to know which phylogeny corresponds to which tool. How the SNV phylogenetic analysis was performed (Supplementary Fig 1)? The bioinformatics method descriptions look sloppy and oversimplified. For instance, the MAFFT and fasttree are just part of the roary pipeline which I assume the author really ran. But the writings looked like the authors run the internal programs independently. Please rewrite these paragraphs in both methods and results.

3. I felt one conclusion of significantly more divergence in *Kingella kingae*, which is defined by > 50 SNVs, is a bit strong without comparing SNV rates from closely-related *Kingella* species. I knew the authors refer to the *Enterococcus* work [13]. However, as the isolates were collected from a diversity of geographical locations, an uncertain amount of diversity due to localized evolution is expected. Why the authors did not compare the phylogenetic clusters with geographical locations? Even if no correlation is found, this would at least indicate no evidence of localized evolution, which is more valuable than the diversity claims.

4. The authors presented several sets of genes present or absent in one particular subgroup (e.g., FrpC absent in endocarditis). At first, this sounds a bit contradictory to the initial findings that no phylogenetic clusters are found associated with each subgroup. I would suggest further classifying these genes vertically or horizontally transferred by double-checking the mobile genetic elements (e.g., IS) in the upstream or downstream of these genes. The evidence of these genes being horizontally transferred would be more consistent with the initial phylogeny analysis.

Minor comments:

A few software used but without properly citations, e.g., roary, fasttree.

Reviewer #2 (Comments for the Author):

In this manuscript, Murik et al. sequenced the genomes of 125 *Kingella kingae* isolates from Israel, Europe, North America, and elsewhere. These *K. kingae* isolates were obtained from healthy carriers and individuals with invasive infections, including bacteremia, endocarditis, and osteoarthritis. The authors then assembled genomes and performed bioinformatic analyses for multilocus sequence typing and to potentially identify genes differentially associated with bacteremia, endocarditis, and osteoarthritis. From this work, Murik et al. placed most of their isolates into existing sequence types, but also identified 15 isolates with novel sequence types. In addition, Murik et al. identified 43 genes that were more likely to be encoded by *K. kingae* isolates from individuals with disease and found that *frpC* was absent in all endocarditis-associated isolates. Overall, the manuscript is well-written, but I have some points of concern to be addressed:

Major

L234-5, 240-2: Is it explicitly known that all *K. kingae* isolates sequenced initially colonized the oropharynx? Is it possible that all *K. kingae* isolates could penetrate the epithelial barrier, but this has not been observed? How do we know that the isolates from healthy carriers were unable to cause bacteremia, endocarditis, or osteoarthritis? How would this affect the interpretation of the gene enrichment analyses?

L214: Can the authors speculate on the roles of the differentially enriched genes in the pathogenesis of *K. kingae* in the discussion?

Figure 1: How does one interpret Panel B? What are the edges representing? Does the distance matter? What is each node? What does it mean for a node to connect to multiple nodes? In Panel C, where is the scale bar for the phylogenetic tree? The colored dots cover the branches. What is the outgroup? Is there bootstrap support for this topology?

Tables S1-3: As far as I can tell, there has not been correction for multiple hypothesis testing for the results presented in Tables S1 through S3. This is especially important as many of the enriched genes are on the borderline for significance.

Minor

P117: Include a supplemental table with all of the genome sequencing statistics for each isolate sequenced in this study

L207: This Chi-square test was not significant. Why were differences pursued?

L226-32: How does the heterogeneity in *K. kingae* compare to other *Neisseriaceae*?

Table 1: Given the size of this table, I would suggest that it should be moved to the Supplemental Information

Figure 3: All panels should be on the same Y-axis scale if they are placed next to each other

Line Comments

L61, 187: Delete "mere"
L61: Instead of "a few genes", give the exact number
L63: Change "...that K. kingae's virulence..." to "...the virulence of K. kingae..."
L64: What constitutes "subtle changes in allelic content" more specifically?
L66: Change "...determinants of K. kingae's invasiveness..." to "...the invasiveness of K. kingae..."
L74: What is HACEK? What is the relevance to this paper?
L74: Change "etiology" to "cause"
L78: Elaborate on what is meant by "portal"
L81: What is "occult bacteremia"?
L87: Delete "exquisite"
L89: Change "the organism" to "this pathogen" or something similar
L93: Delete "would"
P94: Parts of this paragraph are written in active voice and others in passive voice
L95: Why would it be specifically thought that isolates of K. kingae from the endocardium harbor specific genomic determinants? What about the host?
L105-6: Delete "in general"
L106: Delete "in particular"
L116, 124: Change the result subheadings to be more descriptive of the specific findings
L118, 122, 357: Delete the hyphen in "de-novo"
L125: Delete the underscore
L127: Change "exhibited reduced similarity" to "were less similar"
L132: Change "comprised" to "include"
L134: Why could these isolates not be assigned to a specific ST?
L144: Change "...resulting in endocarditis..." to "...cultured from endocarditis patients..."
L155: What is a shell gene? What is a cloud gene?
L157: What is the relevance of the eggNOG-mapper analysis to this paper?
L167, 171, 222: Use a different word than "disregarding"
L170: "...only for genes present in {greater than or equal to}5 isolates." is information for the figure legend
L174: What are the 8 genes?
L180, 499: COG Category Q is for "Secondary Metabolism" not "Secondary Structure"
L186: Change "look for" to "identify"
L195-7: change "are" to "encode". These are genes that encode these proteins.
L203: What statistical test was used here?
L206: Why was this "remarkable"?
L223-5: Wasn't this information already known before the outset of this study based on the previously sequenced genomes?
L239: Replace "sine-qua-non" with a more commonly used phrase
L264-6: What is the relevance of the "non-endocarditis" strains in this context?
L285: What is "vegetations" here?
L290: Delete the hyphen in "pointed-out"
L290-2: Is this finding with fprC consistent with other isolates of NEI K. kingae?
L297: Delete "could"
L300: Change "K. kingae's" to "K. kingae"
L304: Change "childhdod" to "childhood"
L306: Can the authors speculate on what potential vaccine candidates might be used based on their study?
L309-12: Unnecessary information
L312: Delete "Besides"
L371: Capitalize "P-value"

Staff Comments:

Preparing Revision Guidelines

- Point-by-point responses to the issues raised by the reviewers in a file named "Response to Reviewers," NOT IN YOUR COVER LETTER.
- Upload a compare copy of the manuscript (without figures) as a "Marked-Up Manuscript" file.
- Each figure must be uploaded as a separate file, and any multipanel figures must be assembled into one file.

- Manuscript: A .DOC version of the revised manuscript
- Figures: Editable, high-resolution, individual figure files are required at revision, TIFF or EPS files are preferred

Please return the manuscript within 60 days; if you cannot complete the modification within this time period, please contact me. If you do not wish to modify the manuscript and prefer to submit it to another journal, please notify me of your decision immediately so that the manuscript may be formally withdrawn from consideration by Microbiology Spectrum.

Point-by-point responses to the issues raised by the reviewers

Spectrum03895-22: Whole Genomic Sequencing Reveals Differences among *Kingella kingae* Strains from Carriers and Patients with Invasive Infections

Reviewer #1:

Murisk et al. sequenced 125 *Kingella kingae* isolates associated with bacteremia (n=23), osteoarthritis (n=61), and endocarditis (n=18). The authors identified 43 genes possibly discriminative of healthy and invasion, as well as others associated with skeletal system infections, bacteremia, and endocarditis. The FrpC was absent in all endocarditis samples but present in one-third of other invasive samples. In all, the authors provide rich genomic material for further research on *Kingella kingae* infections. However, the manuscript still needs some clarifications on both the results and methods.

1. It looks to me Figure 1(B) and (C) redundantly plotted the cgMLST (or core genes) phylogeny of the 125 isolates just by two distinct software. This confused the audience as it provided no new messages. Instead, I would suggest comparing the cgMLST with the MLST phylogenies, which may at least see the strength and weaknesses of each method.

Reply: Figure 1B presents the relatedness between cgMLST types, but the distances between nodes does not relate to phylogenetic differences. The tree in 1C presents phylogenetic analysis, giving the reader another layer of data, the evolutionary distance between isolates. We believe that both representations are required to show that Endocarditis causing isolates are not related. However, we do agree that it may confuse some of the readers. Now, as suggested by the reviewer, Fig 1C presents the same cgMLST as 1B but color coded to STc (sequence type complex) in order to compare between them.

2. It's difficult to connect the phylogenetic analysis in the results with the methods. It looks like there are three cgMLST-like analyses performed (L137-

151). But only roary and chewBBACA appeared in the methods. And it's hard to know which phylogeny corresponds to which tool. How the SNV phylogenetic analysis was performed (Supplementary Fig 1)? The bioinformatics method descriptions look sloppy and oversimplified. For instance, the MAFFT and fastree are just part of the roary pipeline which I assume the author really ran. But the writings looked like the authors run the internal programs independently. Please rewrite these paragraphs in both methods and results.

Reply: in L137-151 there are now 2 analyses: MLST and cgMLST (we dropped the core genome alignments according to comment #1). The cgMLST was analyzed with chewBACCA as now described more clearly in the Methods. Roary was used to describe the pan- and core- genomes. We removed the sentences involving MAFFT and FastTree (which previously were run independently) as currently their output is not described in the text or figures anymore. All of these changes were updated in both Results and Methods sections.

3. I felt one conclusion of significantly more divergence in *Kingella kingae*, which is defined by > 50 SNVs, is a bit strong without comparing SNV rates from closely-related *Kingella* species. I knew the authors refer to the *Enterococcus* work [13]. However, as the isolates were collected from a diversity of geographical locations, an uncertain amount of diversity due to localized evolution is expected. Why the authors did not compare the phylogenetic clusters with geographical locations? Even if no correlation is found, this would at least indicate no evidence of localized evolution, which is more valuable than the diversity claims.

Reply: *Kingella* species other than *K. kingae* are rare human pathogens from which few clinical isolates are available, and thus, the genomic heterogeneity of these species has not been studied in detail. The 125 *K. kingae* strains included in the present investigation were chosen to represent a diverse sample of the species population. These strains originated in seven countries on three continents, were isolated over a >50-year period, and belonged to 56 different

MLST sequence types (see Table 1). Among strains isolated from the same country, special effort was made to include organisms detected in geographically distant locations and at different times. These precautions avoided including epidemiologically related organisms sharing closely related ancestors and expressing local evolution.

4. The authors presented several sets of genes present or absent in one particular subgroup (e.g., FrpC absent in endocarditis). At first, this sounds a bit contradictory to the initial findings that no phylogenetic clusters are found associated with each subgroup. I would suggest further classifying these genes vertically or horizontally transferred by double-checking the mobile genetic elements (e.g., IS) in the upstream or downstream of these genes. The evidence of these genes being horizontally transferred would be more consistent with the initial phylogeny analysis.

Reply: This is a very interesting comment, and we thank the reviewer for bringing this up. We manually analyzed the regions surrounding the six genes with significant enrichment in isolates from endocarditis patients. None of them (including several kb of flanking regions) mapped to a known IS according to the ISFinder database (<https://isfinder.biotoul.fr/>). Further investigation revealed an inverted repeat flanking the PAS-domain containing gene that may take part in the mechanism of mobile element transfer but no transposase was found in this region. However, since the genome assemblies were produced from single-end short read sequencing data, we are aware that repetitive regions like mobile elements might not be correctly represented in the assemblies.

Minor comments: A few software used but without properly citations, e.g., roary, fastree.

Reply: the missing citations were added.

Reviewer #2:

In this manuscript, Murik et al. sequenced the genomes of 125 *Kingella kingae* isolates from Israel, Europe, North America, and elsewhere. These *K. kingae* isolates were obtained from healthy carriers and individuals with invasive infections, including bacteremia, endocarditis, and osteoarthritis. The authors then assembled genomes and performed bioinformatic analyses for multilocus sequence typing and to potentially identify genes differentially associated with bacteremia, endocarditis, and osteoarthritis. From this work, Murik et al. placed most of their isolates into existing sequence types, but also identified 15 isolates with novel sequence types. In addition, Murik et al. identified 43 genes that were more likely to be encoded by *K. kingae* isolates from individuals with disease and found that *frpC* was absent in all endocarditis-associated isolates. Overall, the manuscript is well-written, but I have some points of concern to be addressed:

Major

L234-5, 240-2: Is it explicitly known that all *K. kingae* isolates sequenced initially colonized the oropharynx? Is it possible that all *K. kingae* isolates could penetrate the epithelial barrier, but this has not been observed? How do we know that the isolates from healthy carriers were unable to cause bacteremia, endocarditis, or osteoarthritis? How would this affect the interpretation of the gene enrichment analyses?

Reply: It has been demonstrated that in children with *K. kingae* disease, strains isolated from the oropharynx are identical to those detected in infected sites, indicating that the colonized epithelium is the source of the bloodstream,

endocardial, joint, and bone invasions, and all invasive isolates had their origin in the colonized upper respiratory mucosal surface. We do not claim that the fact that a *K. kingae* strain was isolated from the oropharynx of a healthy carrier implies, necessarily, that the organism is unable to cause an invasive infection. The species shows a remarkable genomic heterogeneity, similar to other human pathogens and carried organisms exhibit more genomic variability than disease-associated, indicating that not all pharyngeal colonizers are invasive. In a retrospective Israeli study comprising 181 invasive infections, five of 32 (15.6%) PFGE clones (named B, H, K, N, and P) were isolated in 132 (72.9%) patients, implying increased virulence. [Clin Infect Dis 2012; 55:1074-1079]. Remarkably, clones A, C, G, J, M, R, and T, which collectively represented 93 (38.8%) of all strains recovered from 240 healthy carriers in a community survey [Pediatr Infect Dis J 2009; 28:707-710], were isolated in only 4.4% of invasive infections, suggesting a diminished invasive capability. These observations have been confirmed in murine and rat models of invasive *K. kingae* disease, demonstrating that the members of the species show a wide range of virulence, from a severe and rapidly fatal disease in animals inoculated with a strain isolated from a child with osteoarthritis, to no disease when a respiratory isolate was employed [PLoS One 7:e38078. <http://dx.doi.org/10.1371/journal.pone.0038078>]. Because all the strains in the study had initially colonized the oropharynx, the carried group of isolates probably comprised a mixture of organisms with diverse invasive capabilities. This heterogeneity may have blurred the genomic differences between "true mere colonizers" (i.e., colonizing strains unable to cause disease) and invasive organisms isolated from the bloodstream and deep body tissues. Despite this potential attenuation, the comparisons demonstrated genomic differences between organisms isolated from the pharyngeal mucosa and those detected from infected body sites. A new paragraph explaining this point was added to the revised version of the text.

L214: Can the authors speculate on the roles of the differentially enriched genes in the pathogenesis of *K. kingae* in the discussion?

Reply: a new paragraph discussing the hypothetical role of the enriched genes was added to the Discussion.

Figure 1: How does one interpret Panel B? What are the edges representing? Does the distance matter? What is each node? What does it mean for a node to connect to multiple nodes? In Panel C, where is the scale bar for the phylogenetic tree? The colored dots cover the branches. What is the outgroup? Is there bootstrap support for this topology?

Reply: As suggested by Reviewer #1 we removed the phylogenetic tree from Figure 1, and placed it in Figure S1. The tree is unrooted, as the evolutionary trail was not of interest here, only the relationships between isolates. FastTree does not use bootstraps for support value, but SH-test. The values are now added to the tree nodes in the tree and to the Fig S1 legend. Regarding panels B and C in the updated Figure 1 – this is a PhyloViz representation of the cgMLST profiles. Each edge is an isolate. The nodes do not represent evolutionary relationships but rather the distance in the cgMLST profile, or how many alleles differ between any two isolates.

Tables S1-3: As far as I can tell, there has not been correction for multiple hypotheses testing for the results presented in Tables S1 through S3. This is especially important as many of the enriched genes are on the borderline for significance.

Reply: We agree and thank the reviewer for noticing this point. We added adjusted p-values to the tables based on Bonferroni correction and indeed most comparisons are not significant. This is now presented in the tables and described in the Methods.

Minor

P117: Include a supplemental table with all of the genome sequencing statistics for each isolate sequenced in this study

Reply: We added Supplementary Table S4 that presents sequencing and assembly statistics for each isolate.

L207: This Chi-square test was not significant. Why were differences pursued?

Reply: Early in the research, it became apparent that MLST sequence type complex (STC) 23/25 was overrepresented among endocarditis-associated strains, being isolated in 8 of 18 (44%) patients. Thus, we assumed that this STC had an affinity to endocardial tissues. However, further analysis showed that the fraction of STC 23/25 among endocarditis-associated strains did not significantly differ from that found among non-endocardial infections, not supporting specific tropism for the cardiac valves.

L226-32: How does the heterogeneity in *K. kingae* compare to other Neisseriaceae?

Reply: The two most common pathogenic Neisseriaceae, *Neisseria meningitidis* [Nat Rev 2020; 18:84-96] and *Neisseria gonorrhoeae* [Lancet Infect Dis 2020; 20:478–86], exhibit an impressive genomic diversity and continuous emergence of novel strains, as the result of high rates of horizontal gene transfer and the selective pressure exerted by the human immune response.

Table 1: Given the size of this table, I would suggest that it should be moved to the Supplemental Information

Reply: Since Spectrum is an online-only journal we do not see how the dimensions of the table are relevant. Importantly, the table presents the spectrum of the geographic, clinical, chronological and genetic (Sequence Type) variability of the bacteria used in this study. However, if the editor agrees with the

comment, Table 1 can be transferred to the SI. We also have a shorter alternative version of this table, with only N isolate counts for each geographic, clinical and chronological point.

Figure 3: All panels should be on the same Y-axis scale if they are placed next to each other.

Reply: Each panel presents a different set of proteins with large variability in the overall total number of proteins. A large part of the data will not be visible if we scale all panels to the same scale according to the group with the largest N. Therefore, each panel contains its own Y-axis scale which allows the reader to assess the data and the differences between the boxplots.

Line Comments

L61, 187: Delete "mere"

Reply: Done.

L61: Instead of "a few genes", give the exact number

Reply: The study made multiple comparisons between strains isolated from the oropharynx, and those recovered from patients with bacteremia, skeletal system infections, and endocarditis. Each comparison resulted in the identification of multiple genes showing differential distribution. Therefore, we decided to summarize these results as "several genes displayed a significant differential distribution, etc." for the sake of brevity. The actual genes are shown in Supplemental Tables 1, 2, and 3.

L63: Change "...that K. kingae's virulence..." to "...the virulence of K. kingae..."

Reply: Done.

L64: What constitutes "subtle changes in allelic content" more specifically?

Reply: The imprecise word "subtle" was deleted from the revised text.

L66: Change "...determinants of *K. kingae*'s invasiveness..." to "...the invasiveness of *K. kingae*..."

Reply: Done.

L74: What is HACEK? What is the relevance to this paper?

Reply: HACEK is an acronym comprising a group of fastidious facultative anaerobic Gram-negative organisms colonizing the oropharynx and collectively responsible for 5% of all cases of bacterial endocarditis. It comprises *Haemophilus* species, *Aggregatibacter actinomycetemcomitans*, *Cardiobacterium hominis*, *Eikenella corrodens*, and *Kingella* species. The original sentence was deleted from the revised manuscript for brevity.

L74: Change "etiology" to "cause"

Reply: Done.

L78: Elaborate on what is meant by "portal"

Reply: The word "portal" was deleted for the sake of clarity.

L81: What is "occult bacteremia"?

Reply: Bacteremia with no focal infection. This was explained in the revised version of the manuscript (page 16, lines 370-371).

L87: Delete "exquisite"

Reply: Done.

L89: Change "the organism" to "this pathogen" or something similar

Reply: Done.

L93: Delete "would"

Reply: Done.

P94: Parts of this paragraph are written in active voice and others in passive voice

Reply: Done.

L95: Why would it be specifically thought that isolates of *K. kingae* from the endocardium harbor specific genomic determinants? What about the host?

Reply: The contrast between the benign clinical presentation of *K. kingae* skeletal system infections and the severe clinical course of bacterial endocarditis is remarkable. Moreover, the complications and mortality rates of *K. kingae* endocarditis are much higher than those of oral streptococci, comparable to *Staphylococcus aureus*. The referee is correct that the host's factors may account for some of these features. However, it should be noted that *K. kingae* endocarditis frequently affects previously healthy children without predisposing them to medical ailments. The possibility that the local conditions prevalent in the cardiac valves vegetations, which are poorly accessible to antibiotics, neutrophils, antibodies, and other host defense molecules, may contribute to the aggressiveness of the endocardial disease was already discussed in the original submission (original lines 283-290, now on page 13, Lines 309-315).

L105-6: Delete "in general"

Reply: Done.

L106: Delete "in particular"

Reply: Done.

L116, 124: Change the result subheadings to be more descriptive of the specific findings

Reply: Done.

L118, 122, 357: Delete the hyphen in "de-novo"

Reply: Done.

L125: Delete the underscore

Reply: Done.

L127: Change "exhibited reduced similarity" to "were less similar"

Reply: Done.

L132: Change "comprised" to "include"

Reply: Done.

L134: Why could these isolates not be assigned to a specific ST?

Reply: These isolates could not be assigned to a specific complex (not to a sequence type) because they did not differ at no more than one locus from at least one other member of the group (lines 336-337 in the original submission, now on page 17, lines 403-404).

L144: Change "...resulting in endocarditis..." to "...cultured from endocarditis patients..."

Reply: Done.

L155: What is a shell gene? What is a cloud gene?

Reply: A "shell gene", according to Roary tools, is found in 15%-95% of isolate cohort genomes. This was noted in the text: "814 shell genes found in 15%-95%

isolates" (Line 170). A "cloud gene", according to Roary tools, is found in <15% of the isolates. This definition is in the text right after the aforementioned sentence.

L157: What is the relevance of the eggno-mapper analysis to this paper?

Reply: We used prokka as the first tier annotation tool for all assemblies. Since a large portion of the predicted genes were annotated as "putative proteins" we used the eggno mapper in order to identify more detailed functional annotation or evolutionary evidence for some of these unannotated genes.

L167, 171, 222: Use a different word than "disregarding"

Reply: Done

L170: "...only for genes present in {greater than or equal to}5 isolates." is information for the figure legend

Reply: It is mentioned in the figure legend.

L174: What are the 8 genes?

Reply: The cut-off for inclusion in the Venn diagram is ≥ 5 isolates from each clinical group. All 8 genes found to pass this criteria only in endocarditis causing isolates were also found (up to 4 times) in the other groups, and none of them were significantly enriched in any of the clinical outcome groups. For this reason we did not discuss the genes any further. Five of the 8 genes do not have any hit in the Swissprot database, 2 are similar to probable type-III restriction enzymes, and 1 is similar to an iron regulated protein.

April 26, 2023

Prof. Pablo Yagupsky
Ben-Gurion University of the Negev
Clinical Microbiology Laboratory
Shderot Reger
Beer-Sheva
Israel

Re: Spectrum03895-22R1 (Whole Genomic Sequencing Reveals Differences among *Kingella kingae* Strains from Carriers and Patients with Invasive Infections)

Dear Prof. Pablo Yagupsky:

Your manuscript has been accepted, and I am forwarding it to the ASM Journals Department for publication. You will be notified when your proofs are ready to be viewed.

Sincerely,

Po-Yu Liu
Editor, Microbiology Spectrum
